# Batch Manufacturing of Split-Actuator Micro Air Vehicle Based on Monolithic Processing Technology

**DOI:** 10.3390/mi12101270

**Published:** 2021-10-18

**Authors:** Xiang Lu, Chengxiang Wang, Kun Lu, Xiang Xi, Yulie Wu, Xuezhong Wu, Dingbang Xiao

**Affiliations:** College of Intelligent Science, National University of Defense Technology, Changsha 410073, China; luxiang@nudt.edu.cn (X.L.); cx.wangnudt@nudt.edu.cn (C.W.); lukun@nudt.edu.cn (K.L.); xixiang@nudt.edu.cn (X.X.); ylwu@nudt.edu.cn (Y.W.); xzwu@nudt.edu.cn (X.W.)

**Keywords:** microrobots, monolithic process, flapping-wing, air vehicle, batch manufacture

## Abstract

Microrobots have a wide range of applications. The rigid–flexible composite stereoscopic technology based on ultraviolet laser cutting technology is primarily researched for the design and manufacture of microrobots and has been used to fabricate microscale motion mechanisms and robots. This paper introduces a monolithic processing technology based on the rigid–flexible composite stereoscopic process. Based on this process, a split-actuator micro flapping-wing air vehicle with a size of 15 mm × 2.5 mm × 30 mm was designed. We proposed a batch manufacturing method capable of processing multiple micro air vehicles at the same time. The main structure of 22 flapping-wing micro air vehicles can be processed at the same time within the processing range of the composite sheet with an area of 80 mm × 80 mm, and the processing effect is good.

## 1. Introduction

With the rapid development of computer-based integration technology, material science, aerodynamics, and micro-nano manufacturing, small bionic microrobots have aroused the interest of research institutes and scholars worldwide. These robots, which have the characteristics of solid concealment, flexibility, mobility, and economy, can be widely used in surveillance, search, and rescue utility in dangerous, complex, and unknown environments, such as ruins [1,2,3,4]. Researchers have focused on the innovation of the robot structure, the optimization of the manufacturing process, the invention of the energy supply and driving mode, and the optimization of the control perception during the flapping-wing micro air vehicle research process.

Processing and manufacturing macrosize robots usually rely on traditional mechanical manufacturing methods such as turning, milling, planing, grinding, drilling, etc. However, because the full size or feature size of microrobots has reached the micro-nano level, these traditional mechanical manufacturing methods are no longer suitable for micro- and nanorobots. As a result, some new processing techniques for micro- and nanorobot fabrication have been developed in recent decades, such as MEMS etching [5,6], deposition [7], 3D printing [8,9,10], laser cutting [11,12,13], pop-up process [14,15,16,17,18], and so on. Three-dimensional (3D) printing is one of the most promising processing methods. This method can quickly process complex microstructures, but due to technical limitations, printing materials are limited, structural strength is fragile, and manufacturing efficiency is low. MEMS etching and deposition technology can process micro-nanoscale structures with high processing accuracy, which will be useful in the future for nanorobot structures. However, it also faced a number of challenges, including fragile silicon-based materials, stringent processing conditions, and high processing costs. The primary application of ultraviolet laser cutting is in the 2D processing of thin-layer materials.

In this paper, we first designed a miniature flapping-wing air vehicle with a split actuator and proposed an integrated processing technology for the air vehicle based on the rigid–flexible composite 3D process. Then, we optimized the process to realize the batch manufacturing of the air vehicle. Finally, we carried out the actuation test of the air vehicle.

## 2. Materials and Methods

### 2.1. Design of Separate-Driven Micro Air Vehicle

Microscale mobile robots are obtained by bionics, whose mechanical structures are almost identical to macroscopic robots, including driving mechanisms, transmission mechanisms, motion mechanisms, frame, energy supply, and sensing equipment. Batch machining methods based on monolithic processing technology can be used for batch machining such microrobots. Specific machining methods differ for robots with different motion types and structures, but the overall machining concept is consistent. Flapping-wing micro air vehicles were inspired by dipteran insects, which have a trunk, wings, flight muscles, and transmission bones [19,20]. Such mechanisms should be included in the flapping-wing micro air vehicle. Figure 1 depicts a 3D model of a flapping-wing micro air vehicle. The wingspan of the flapping-wing micro air vehicle is about 30 mm, and the fuselage size is about 15 mm × 2.5 mm × 30 mm.

The driving mechanism of this flapping-wing micro air vehicle uses the piezoelectric ceramic bimorph bending actuator [21], which was designed by the team of Professor R J Wood of Harvard University [22,23]. The actuator was made of PZT−5H piezoelectric ceramic sheets, carbon fiber, and glass fiber sheets, as shown in Figure 2a. The movement of insects’ wings in flight is mainly composed of two parts: torsional movement along the leading edge of the wings and reciprocating flapping movement with a high angle of attack. They are actuated by the flight muscles of the chest cavity with multiple rotational degrees of freedom, and the generation of the lift force is realized under the joint action of the two movements. We use the terms “upstroke” and “downstroke” to describe the flapping movement of the wings from belly to back and back to belly. A four-bar linkage mechanism is designed as the transmission mechanism of the flapping-wing micro air vehicle, inspired by the system of the chest cavity of the bee shown in Figure 2b. The hinge mechanism is capable of converting minor deflections into visible angle changes. Figure 2c depicts the plane mechanisms design. The transmission mechanism is composed of two symmetrical arrangements on both sides of the four-bar linkage. The middle of the symmetrical linkage is the actuate input. The two linkage mechanisms can amplify the input actuate displacement into the flapping motion of the wings. This transmission mechanism is a single-input dual-output type, so the flapping motion of the two wings is always symmetrical, and it can only generate upward lift. It cannot create forces and moments in other directions through asymmetric wing motion. As a result, the air vehicle can only perform a single freedom of movement. The structure is optimized to achieve the spherical four-bar linkage transmission mechanism depicted in Figure 2d, the machine has a split-actuator design, and the left and right structures are symmetrical. To achieve decoupling of the movement of the two wings, one driver corresponds to the drive control of one wing. The actuate is placed perpendicular to the transmission mechanism, two actuators can be placed symmetrically, and space can be saved. The two wings can move independently. The frequency, flapping amplitude, and initial offset position of the two wings can be adjusted asymmetrically to achieve the combined force of lift and drag in other directions, thereby generating pitch, pitch, and yaw moments. It can make the air vehicle move in multiple degrees of freedom.

The transmission ratio of the transmission mechanism is calculated as follows. Each spherical four-bar linkage mechanism consists of three flexible hinge joints and four rigid linkages (a U-shaped rod), respectively, $L_{0}$, $L_{1}$, $L_{2}$, $L_{3}$, and $L_{4}$. As shown in Figure 2e, the input given by the piezoelectric drive mechanism is $\alpha_{in}$, the output of the corresponding wings is $\theta_{out}$, and the deflection angles corresponding to the three flexible hinges are $\theta_{1}$, $\theta_{2}$, and $\theta_{3}$. Through geometric plane analysis, the following geometric relationships exist:(1)$$\{\begin{array}{c} \theta_{3}=\theta_{1}+\theta_{2}\\ L_{1}\cos\theta_{1}+L_{3}\sin\theta_{3}-\alpha_{in,1}=L_{1}\\ L_{3}\cos\theta_{3}-L_{1}\sin\theta_{1}=L_{3}\\ \theta_{out}=\theta_{3} \end{array}$$

The above four geometric relation equations are combined to obtain the following results:(2)$$\{\begin{array}{c} \theta_{1}=\sin^{-1}\left(\frac{L_{3}\left(\cos\theta_{out}-1\right)}{L_{1}}\right)\\ \theta_{2}=\theta_{out}-\sin^{-1}\left(\frac{L_{3}\left(\cos\theta_{out}-1\right)}{L_{1}}\right)\\ \alpha_{in,1}=\sqrt{L_{1}^{2}-L_{3}^{2}{\left(\cos\theta_{out}-1\right)}^{2}}-L_{1}+L_{3}\sin\theta_{out} \end{array}$$

The transmission ratio is the change of the output angle of the wings $\theta_{out}$ and the input displacement of the piezoelectric actuator $\alpha_{in}$, which the average transmission ratio can express:(3)$$\begin{array}{l} T=\frac{\theta_{out}}{\alpha_{in}}\approx \frac{\theta_{out,\max}-\theta_{out,\min}}{\alpha_{in,\max}-\alpha_{in,\min}}\\ =\frac{\theta_{out,\max}-\theta_{out,\min}}{L_{3}(\sin\theta_{out,\max}-\sin\theta_{out,\min})+(\sqrt{L_{1}^{2}-L_{3}^{2}{\left(\cos\theta_{out,\max}-1\right)}^{2}}-\sqrt{L_{1}^{2}-L_{3}^{2}{\left(\cos\theta_{out,\min}-1\right)}^{2}})} \end{array}$$

Since $\theta_{out,\max}=-\theta_{out,\min}$, Formula (3) can be simplified to:(4)$$T\approx \frac{\theta_{out,\max}-(-\theta_{out,\max})}{L_{3}(\sin\theta_{out,\max}+\sin\theta_{out,\max})}=\frac{\theta_{out,\max}}{L_{3}\sin\theta_{out,\max}}$$

The limit value of the output displacement of the piezoelectric driver is ±300 μm, and the flapping amplitude of the wings is expected to reach ±60°, so the designed transmission ratio should get $T=\frac{\left(\pi /3\right)\mathrm{rad}}{0.3\mathrm{mm}}=3.49\mathrm{rad}/\mathrm{mm}$. When the flapping angle of the wings comes 60°, the transmission ratio is:(5)$$T=\frac{\theta_{out,\max}}{L_{3}\sin\theta_{out,\max}}=\frac{1.2}{L_{3}}=3.49\;\mathrm{rad}/\mathrm{mm}$$

The transmission mechanism was calculated to be approximately equal to 1.2/L_3_ [24]. Through reasonable arrangement and calculation, the specific dimensions of the connecting rods of the driver and the transmission mechanism can be obtained, as shown in Table 1.

### 2.2. Monolithic Manufacturing Technology of Flapping-Wing Micro Air Vehicle

The main structure of the flapping-wing micro air vehicle was processed by a monolithic manufacturing technology based on the rigid–flexible composite stereoscopic process [24]. The whole idea of the process is shown in Figure 3. The first step is to perform 3D modeling of the structure, the second step is to use layered 2D plane processing, and finally, the form processed by the 2D plane is converted into a 3D entity by the designed method. The process is distinguished by the creation of flexure mechanisms and assembly creases by laminating alternating rigid and compliant laser processing materials, followed by subsequent processing to release hinge structure. In the monolithic processing technology of this flapping-wing micro vehicle, the primary system composed of the fuselage, transmission mechanism, and passive hinges of the wings is made by an integral, three-dimensional, rigid–flexible composite process.

The rigid–flexible composite stereoscopic process, illustrated in Figure 4, consists of the following steps:

The carbon fiberboard and resin glue were cured; laser etching was used to etch the designed hinge groove on the carbon fiberboard and the cured resin glue layer;The etched plates are aligned and stacked in order and then cured;The outer contour of the cured rigid–flexible composite material layer was released;The released flat rigid–flexible composite material layer was folded according to the designed structure’s fixed-body deformation mechanics.

**Figure 4 micromachines-12-01270-f004:**
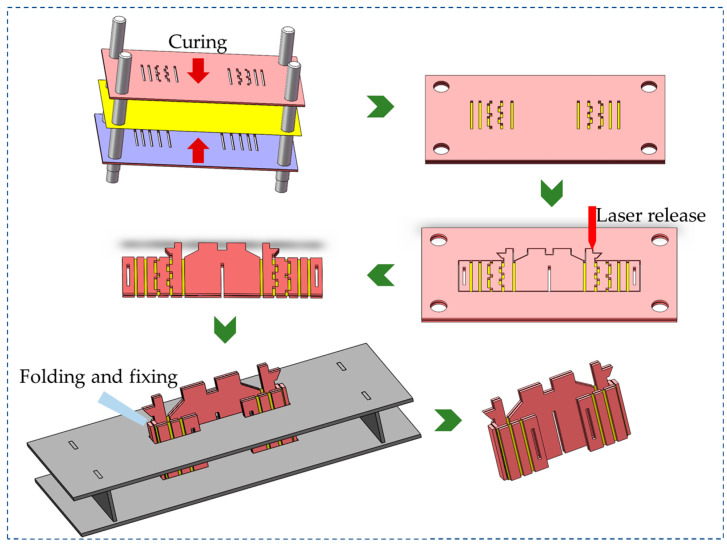
The rigid–flexible composite stereoscopic process.

The actuators and wings adopt vacuum bag curing technology and are super obtained by combining fast laser release processing methods. The vacuum bag equipment comprises a sealed cavity composed of an aluminum alloy cavity, a rubber sealing ring, and a flexible PI film, and an external air pipe is connected, as shown in Figure 5. When the vacuum bag works, the mechanical pump evacuates the sealed cavity composed of aluminum alloy and flexible film through the air pipe to give uniform and constant pressure to the objects placed in the cavity. There are two main types of gluing layer: in the piezoelectric actuator, it is a carbon fiber prepreg, which plays the roles of adhesion and conduction; in the main structure of the micro air vehicle, it is the thermoplastic polyurethanes (TPU), which mainly plays the role of adhesion.

The piezoelectric actuation mechanism adopts the process of combining composite material vacuum bag technology and laser release [24]. Figure 6a depicts its unique structure. The process begins with laser cutting, followed by shape cutting of the piezoelectric sheet and each layer of the support material, and then proceeds in sequence. Curing with the vacuum bag, followed by the release of the actuator. The wings are a rigid–flexible composite structure made up of a rigid vein skeleton and a flexible film. The processing process is similar to that of the actuator, as shown in Figure 6b, the solid vein skeleton is first cut, then the flexible film and the vein skeleton are combined and cured, and finally, the release of the wings is performed.

The designed planar composite sheet structure is processed by adopting the processing method of rigid–flexible composite three-dimensional structure, as shown in Figure 7a. Then, the processed composite board is folded and fixed according to the hinge crease and folding angle, as shown in the figure. Finally, the main structure shown in Figure 7b can be obtained

Finally, the processed drive mechanism, wings, and primary structure are assembled and fixed in the order shown in Figure 8, and a complete prototype of a flap-ping-wing air vehicle can be obtained. The assembly sequence is as follows:

Folding and fixing of the spherical four-bar linkage transmission mechanism;Assemble and fix the wings on the slot under the passive hinge;Folding fixed wings and passive hinges;Assemble the drive mechanism in the slot of the drive mechanism;Fold the body and fix the drive mechanism to complete the assembly of the prototype.

**Figure 8 micromachines-12-01270-f008:**
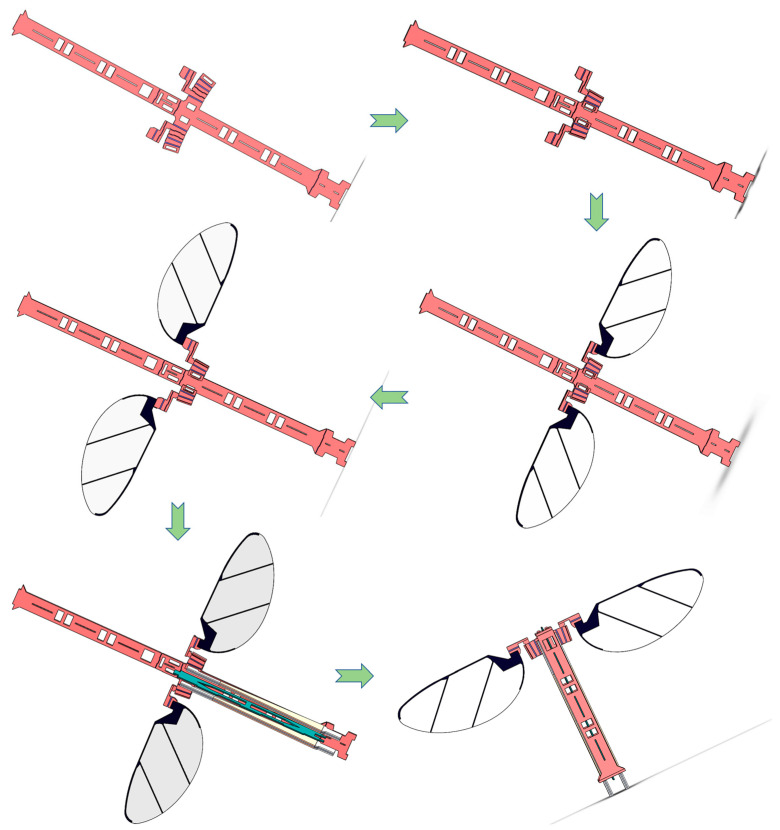
The complete assembly process of integrated manufacturing of flapping-wing air vehicles.

The materials selected for processing the micro flapping wing air vehicle are a 30 μm thick carbon fiber prepreg, a 70 μm thick unidirectional carbon fiber sheet, a 100 μm thick woven carbon fiber sheet, a 100 μm thick Al_2_O_3_ hard sheet, 127 μm thick PZT−5H piezoelectric ceramics, 1.5 μm, and an 8 μm thick polyimide film.

## 3. Results and Discussion

### 3.1. Batch Production of Micro Air Vehicle Prototypes Based on Monolithic Manufacturing

A batch processing method of micron-scale mobile robots’ production is developed by research on monolithic technology. The whole idea of this method is shown in Figure 9a. The repeated structures are rationally distributed on a single processing drawing. A single clamping process can manufacture several structures, which can significantly improve the processing efficiency, save processing costs, and ensure the consistency of processed products.

The standard of alignment and positioning of the sheet is higher with the arrayed machining method, and adding reference holes can enhance the accuracy of the alignment and positioning. It is easier to bend during heating and curing to cooling because the material area used for array processing is several times that of conventional processing, and the thermal expansion coefficient of the rigid sheet and glue layer is inconsistent. After curing, the layer structure can be restored to its original shape by tearing off the film that was initially covered under the glue layer, then recovering the coated paper and solidifying it again, as shown in Figure 9b.

In the production process of the standard laminate layer, due to several material types and the inconsistent coefficient of thermal expansion of the materials, generating sizable internal stress is easy, which leads to severe warpage of the plate. The stress relief holes on the plate are designed and processed before it is cured to form a standard laminate, which can reduce the influence of internal stress. The cured standard laminate without stress relief holes suffered severe warpage, as shown in Figure 9c. When the stress relief holes were added, the cured standard laminates warpage was significantly reduced without affecting the subsequent processing.

The batch manufacturing of flapping-wing micro air vehicles can be realized with the monolithic manufacturing method, reasonable layout of processing drawings, and stress treatment of plates. As shown in Figure 10, the main structures of 22 flapping-wing micro air vehicles can be arranged in processing drawings through design on the composite sheet processing range of 80 mm × 80 mm, and the main structures of multiple air vehicles can be obtained by single processing. Combining the array processing of wings and piezoelectric actuators, multiple flapping-wing micro air vehicles can be manufactured at one time, as shown in Figure 11.

### 3.2. Actuate Test of Flapping-Wing Micro Air Vehicle

The flapping-wing micro air vehicle is driven by piezoelectricity, which needs a higher actuation voltage. In this test, a commercial piezoelectric ceramic controller is used to provide energy to the air vehicle. The controller can output a voltage with an amplitude range of 0–999 V and a frequency range of 0–1500 Hz. For actuation testing, a device was chosen from the batch-processed flapping-wing micro air vehicle.

First, a single-sided wing actuation test was performed, feeding the left-wing actuator a sine drive signal with a peak-to-peak value of 300 V, a bias of 150 V, and a frequency of 50 Hz, observing flapping amplitude on the left side of the wings by about ±65° and a frequency of 50 Hz reciprocating flapping. Figure 12 shows the left wing from the largest positive angle to the largest negative angle of half a flapping movement cycle.

Then, a simultaneous actuation test of both wings was carried out to give both sides, and a sinusoidal drive signal with an actuate peak-to-peak value of 300 V, a bias of 150 V, and a frequency of 80 Hz was given. The wings’ reciprocating flapping motion of 80 Hz frequency was observed with a flapping amplitude of ±45°. Figure 13 shows the flapping action of a flapping-wing air vehicle for half a movement cycle. Due to machining error, the movement of the left and right wings is not entirely symmetrical.

Finally, the two wings were driven asymmetrically. One wing actuator was given a sinusoidal drive signal with a peak-to-peak value of 300 V, a bias of 150 V, and a frequency of 80 Hz. A sinusoidal drive signal with a peak-to-peak value of 100 V, a bias of 50 V, and a frequency of 80 Hz was applied to the other wing actuator. The two wings were discovered to have the same frequency of 80 Hz but different flapping amplitudes. The larger side flapped its wings with an amplitude of ±45°, and the smaller side flapped its wings with an amplitude of ±10°, as shown in Figure 14.

## 4. Conclusions

In this paper, the new processing technology of the microscale mobile robot, the rigid–flexible composites stereoscopic technology, is investigated. This process was inspired by the stereospecific book, which can process and realize a flexible hinge mechanism well. Based on this process, a split-actuator micro flapping-wing air vehicle with dimensions of 15 mm × 2.5 mm × 30 mm was designed. It also proposed monolithic processing based on rigid–flexible composite stereoscopic technology, which can improve processing efficiency while decreasing processing error and complexity. A new array processing technology is proposed that can be used to batch micron-scale movable hinges and mobile robots. Furthermore, this method can improve processing efficiency, reduce processing costs, and ensure product consistency. Moreover, it is capable of small-batch manufacturing. The main structure of 22 flapping-wing micro air vehicles can be processed at once within the processing range of the composite sheet with an area of 80 mm × 80 mm, and the processing effect is good. The successful application of array processing methods on flapping-wing micro air vehicles also provides an economical and feasible reference method for subsequent batch processing and manufacturing micron-level mechanisms and mobile robots.

At present, the processing technology of the microscale mobile robot is relatively mature, but some outstanding problems need to be perfected. One is to realize the minimization of the thermal effect of laser cutting in the process; another is to select lighter and stronger materials for micro-nano robots. These will be the focus of future work.

## Figures and Tables

**Figure 1 micromachines-12-01270-f001:**
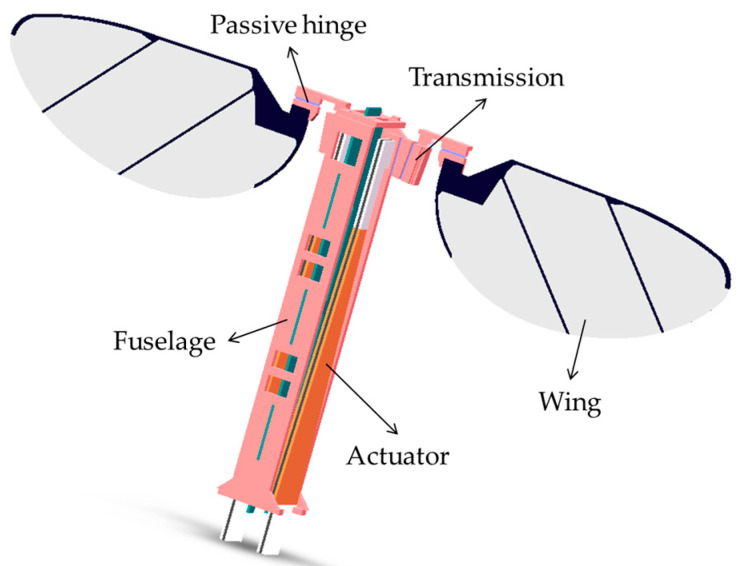
Three-dimensional (3D) model of a flapping-wing micro air vehicle.

**Figure 2 micromachines-12-01270-f002:**
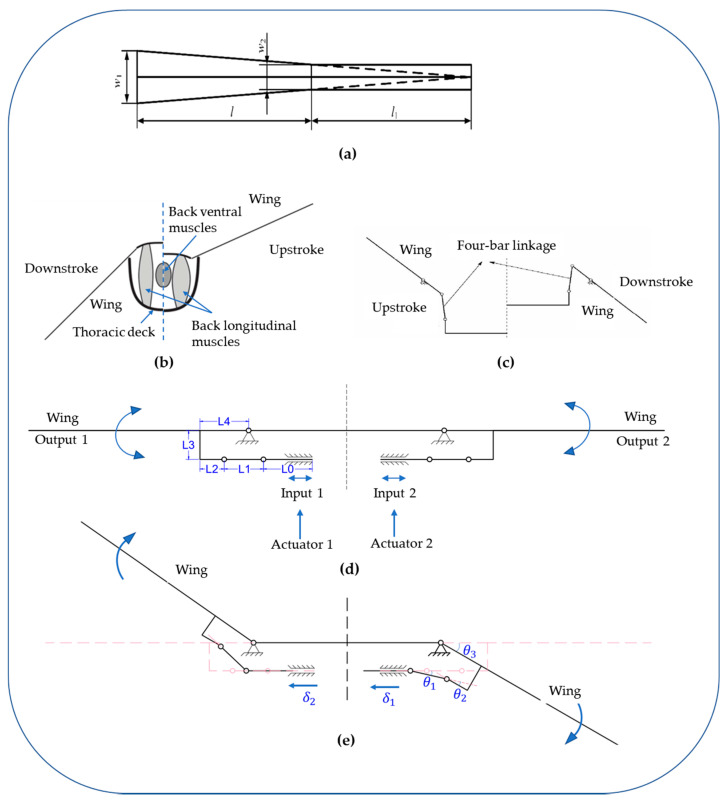
Design of air vehicle mechanism: (**a**) the plane structure of the actuator; (**b**) the structure of the chest cavity of the bee; (**c**) the four-bar linkage mechanism; (**d**) the spherical four-bar linkage transmission mechanism; (**e**) calculation of the transmission ratio of the transmission mechanism.

**Figure 3 micromachines-12-01270-f003:**
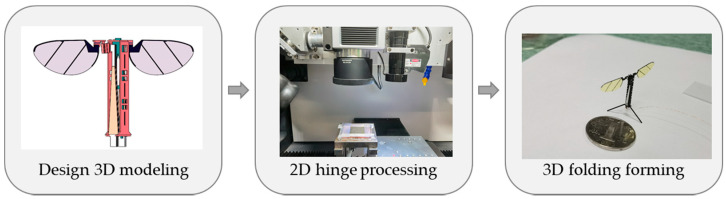
The whole manufacturing idea of the flapping-wing micro air vehicle.

**Figure 5 micromachines-12-01270-f005:**
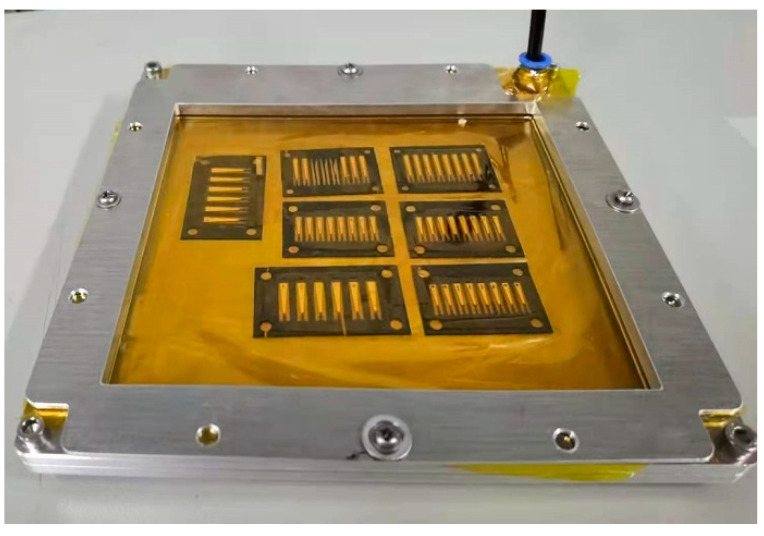
The vacuum bag.

**Figure 6 micromachines-12-01270-f006:**
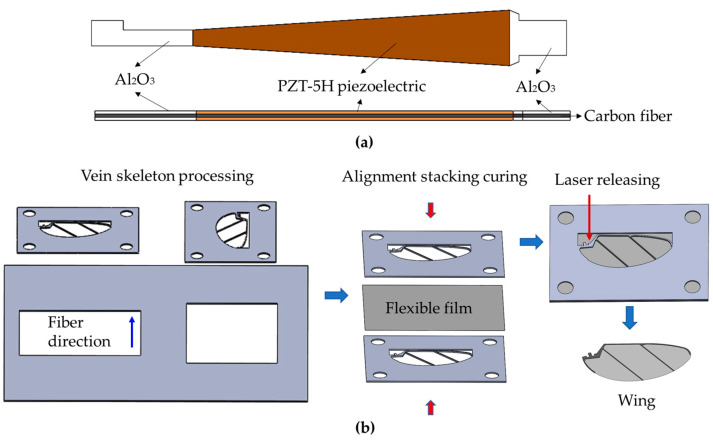
Actuator and wing manufacture: (**a**) specific structure of actuator; (**b**) the manufacturing process of the wings.

**Figure 7 micromachines-12-01270-f007:**
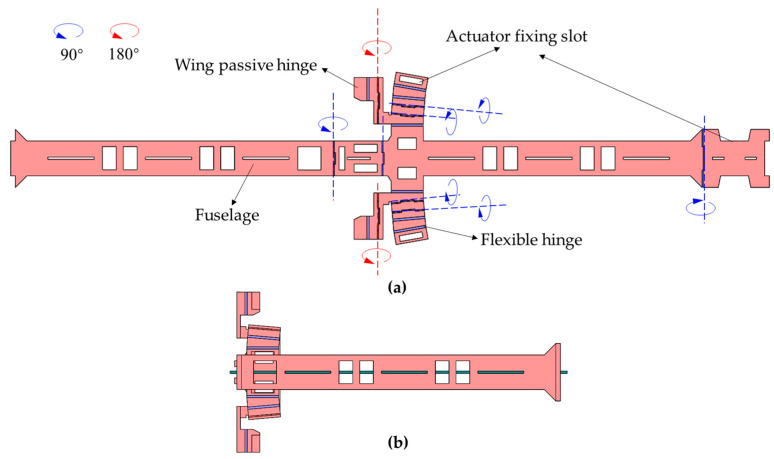
Monolithic manufacturing method of air vehicle main structure: (**a**) folding and shaping method; (**b**) folded main structure.

**Figure 9 micromachines-12-01270-f009:**
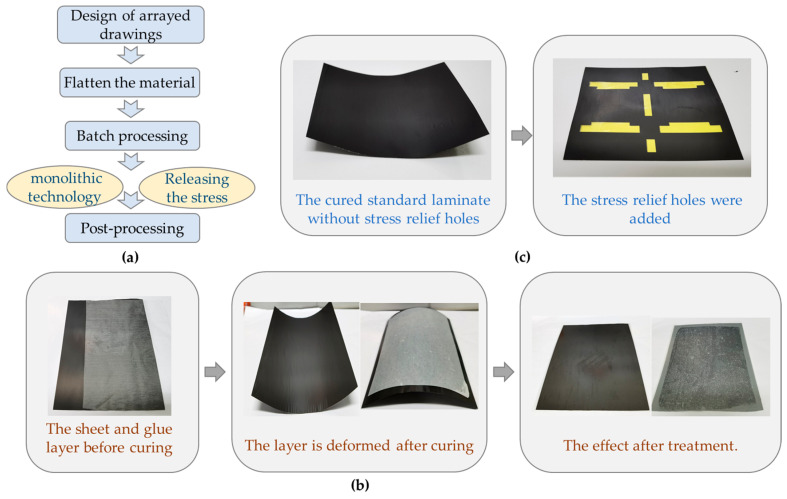
Batch processing method based on monolithic technique: (**a**) the whole idea of batch processing; (**b**) treatment of deformation of material layer; (**c**) standard laminate layer with stress relief holes.

**Figure 10 micromachines-12-01270-f010:**
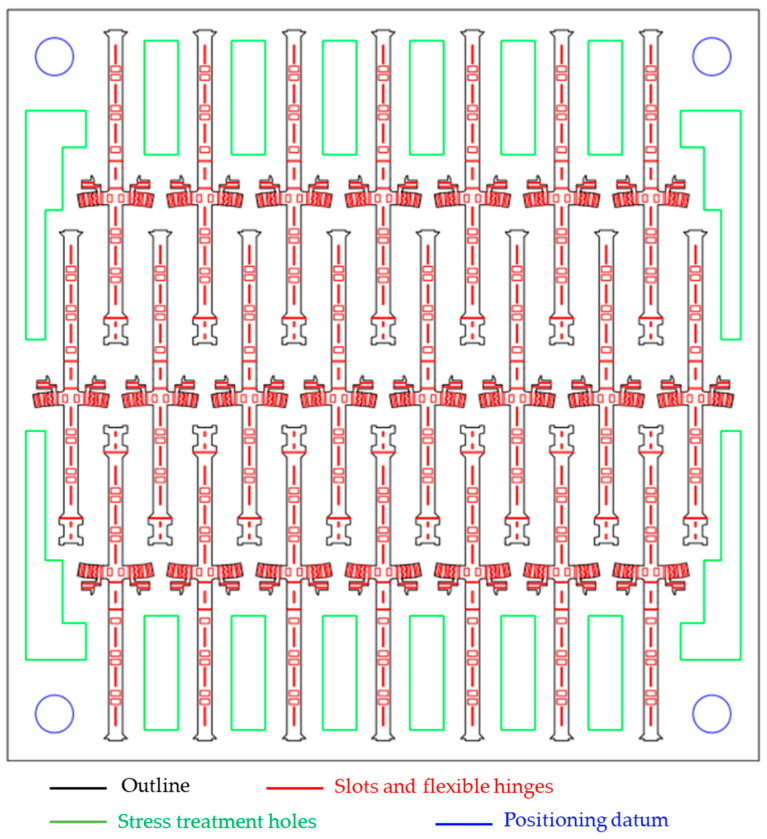
Design drawings for batch processing with a size of 80 mm × 80 mm.

**Figure 11 micromachines-12-01270-f011:**
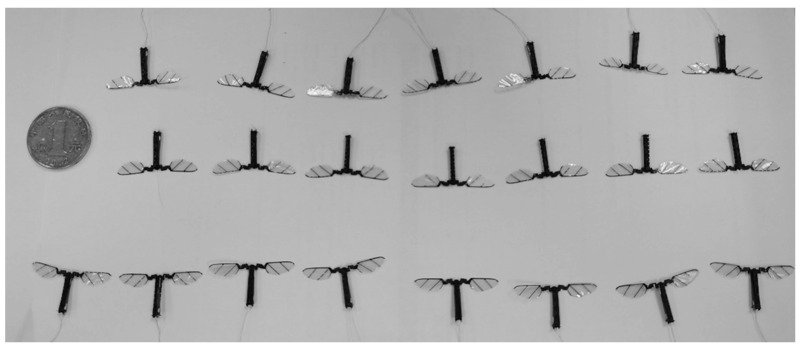
The flapping-wing air vehicle was obtained by batch production.

**Figure 12 micromachines-12-01270-f012:**
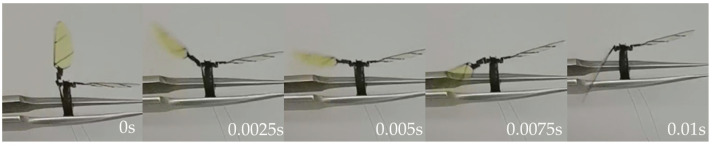
Single-sided wing actuation test.

**Figure 13 micromachines-12-01270-f013:**
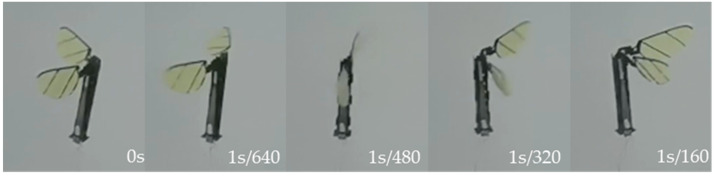
The simultaneous actuation test of both wings.

**Figure 14 micromachines-12-01270-f014:**
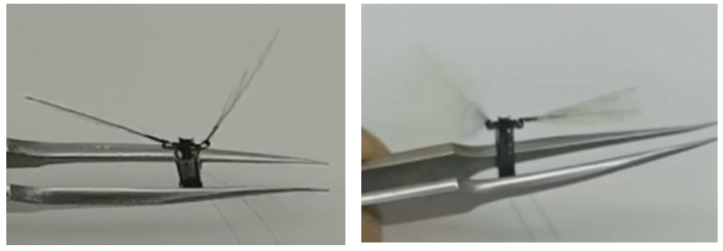
The two wings were driven asymmetrically.

**Table 1 micromachines-12-01270-t001:** The specific dimensions of the actuator and transmission mechanism.

Structural Part	l	$w_{1}$	$w_{2}$	$l_{1}$	*L* _1_	*L* _2_	*L* _3_	*L* _4_
**Size (mm)**	9	1.75	0.5	3.6	0.45	0.25	0.3	0.4

## Data Availability

Data are contained within the article.

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
