# Peer review of "Batch Manufacturing of Split-Actuator Micro Air Vehicle Based on Monolithic Processing Technology"

_micromachines, 2021, doi:10.3390/mi12101270_

Round 1

Reviewer 1 Report

The manuscript presents a monolithic processing technology 
based on the rigid-flexible composite stereoscopic process to produce
micro flapping-wing air vehicle with a size of 15 mm * 2.5 mm * 30 mm. The authors proposed 
a batch manufacturing method to produce multiple micro air vehicles at once. 

1. The English text has minor errors which have to be proofread once more

time.

2. I would like to ask the authors to describe the function of the flying mechanism more precisely, especially Figs. 2 B), c) and d) need more textual explanation.

3. Have the flying mechanism already flied for at least for few parts of the second? This would be fine to mention and make the short movie to prove it. This is the only thing which is missing to have a good paper.

Reviewer 2 Report

The paper ” Batch manufacturing of split actuator micro air vehicle based on monolithic processing technology” by Lu, etc. report a fabrication method for producing a batch of micro sized vehicles, composed of carbon fiber sheet, Al frame and piezo actuator. The design and fabrication have been explained in good details, and the wing motion driven by controlled voltage has been briefly characterized.

The story flow is quite good, however, many typos or writing mistakes exist. I would urge the authors to pay attention. For instance,

Abstract, to fabricates

P1, “and” artificial intelligence technology; of dangerous; has restricted printing material; “However, it also faced various challenges, fragile silicon-based materials”

P2, machining such micro-

P4, Folding and

P6, sin-gle

Section 3.2, the sentences are grammarly inappropriate.

I would suggest the authors to give more details about the manufacture, for example, the type of vacuum bag, and the gluing layer.

P8 “Due to machining error, the movement of the left and right wings is not entirely symmetrical.” Any data to show the machining error, wing actuation differences between two sides?

Also, the reference formats are inconsistent.

Round 2

Reviewer 1 Report

I have no more suggestions!

Reviewer 2 Report

I am fine with this successful revision.